# Towards a More Rigorous Science of Blindspot Discovery in Image Models

## Abstract

A growing body of work studies Blindspot Discovery Methods (BDMs): methods for finding semantically meaningful subsets of the data where an image classifier performs significantly worse, without making strong assumptions. Motivated by observed gaps in prior work, we introduce a new framework for evaluating BDMs, `SpotCheck`, that uses synthetic image datasets to train models with known blindspots and a new BDM, `PlaneSpot`, that uses a 2D image representation. We use `SpotCheck` to run controlled experiments that identify factors that influence BDM performance (*e.g.,* the number of blindspot in a model) and show that `PlaneSpot` outperforms existing BDMs. Importantly, we validate these findings using real data. Overall, we hope that the methodology and analyses presented in this work will serve as a guide for future work on blindspot discovery.

## 1 Introduction

A growing body of work has found that models with high test performance can still make systemic errors, which occur when the model performs significantly worse on a semantically meaningful subset of the data (Buolamwini & Gebru, 2018; Chung et al., 2019; Oakden-Rayner et al., 2020; Singla et al., 2021; Ribeiro & Lundberg, 2022). For example, past works have demonstrated that models trained to diagnose skin cancer from dermoscopic images sometimes rely on spurious artifacts (*e.g.,* surgical skin markers that some dermatologists use to mark lesions); consequently, they have different performance on images with or without those spurious artifacts (Winkler et al., 2019; Mahmood et al., 2021). More broadly, finding systemic errors can help us detect algorithmic bias (Buolamwini & Gebru, 2018) or sensitivity to distribution shifts (Sagawa et al., 2020; Singh et al., 2020).

In this work, we focus on what we call the *blindspot discovery* problem, which is the problem of finding an image classification model's systemic errors[1] without making many of the assumptions considered in related works (*e.g.,* we do not assume access to metadata to define semantically meaningful subsets of the data, tools to produce counterfactual images, a specific model structure or training process, or a human in the loop). We call methods for addressing this problem Blindspot Discovery Methods (BDMs) (*e.g.,* Kim et al., 2019; Sohoni et al., 2020; Singla et al., 2021; d'Eon et al., 2021; Eyuboglu et al., 2022).

We note that blindspot discovery is an emerging research area and that there has been more emphasis on developing BDMs than on formalizing the problem itself. Consequently, we propose a problem formalization, summarize different approaches for evaluating BDMs, and summarize several high-level design choices made by BDMs. When we do this, we observe the following two gaps. First, existing evaluations are based on an incomplete knowledge of the model's blindspots, which limits the types of measurements and claims they can make. Second, dimensionality reduction is a relatively underexplored aspect of BDM design.

Motivated by these gaps in prior work, we propose a new evaluation framework, `SpotCheck`, and a new BDM, `PlaneSpot`. `SpotCheck` is a *synthetic* evaluation framework for BDMs that gives us complete knowledge of the model's blindspots and allows us to identify factors that influence BDM

---

[1]In past work, "systemic errors" have also been called "failure modes" or "hidden stratification." We introduce "blindspot" to mean the same thing and use it make it clear when we are specifically discussing blindspot discovery.

performance. Additionally, we refine the metrics used by past evaluations. `PlaneSpot` is a simple BDM that finds blindspots using a 2D image representation.

We use `SpotCheck` to run controlled experiments that identify factors that influence BDM performance (*e.g.,* the number of blindspot in a model) and show that `PlaneSpot` outperforms existing BDMs. We run additional semi-controlled experiments using the COCO dataset (Lin et al., 2014) and find that these trends discovered using `SpotCheck` generalize to real image data. Overall, we hope that the methodology and analyses presented in this work will help facilitate a more rigorous science of blindspot discovery.

## 2 BACKGROUND

In this section, we formalize the problem of *blindspot discovery* for image classification. We then discuss general approaches for evaluating the Blindspot Discovery Methods (BDMs) designed to address this problem as well as high-level design choices made by BDMs.

**Problem Definition.** The broad goal of finding systematic errors has been studied across a range of problem statements and method assumptions. Some common assumptions are:

- Access to metadata help define coherent subsets of the data (*e.g.,* Kim et al., 2018; Buolamwini & Gebru, 2018; Singh et al., 2020).
- The ability to produce counterfactual images (*e.g.,* Shetty et al., 2019; Singla et al., 2020; Xiao et al., 2021; Leclerc et al., 2021; Bharadhwaj et al., 2021; Plumb et al., 2022).
- A specific structure for the model we are analyzing (*e.g.,* Alvarez-Melis & Jaakkola, 2018; Koh et al., 2020) or for training process used to learn the model's parameters (*e.g.,* Wong et al., 2021).
- A human-in-the loop, either through an interactive interface (*e.g.,* Cabrera et al., 2019; Balayn et al., 2022) or by inspecting explanations (*e.g.,* Yeh et al., 2020; Adebayo et al., 2022).

While appropriate at times, these assumptions all restrict the applicability of their respective methods. For example, consider assuming access to metadata to help define coherent subsets of the data. To start, this metadata is much less common in applied settings than it is for common ML benchmarks. Further, the efficacy of methods that rely on this metadata is limited by the quantity and relevance of this metadata; in general, efficiently collecting large quantities of relevant metadata is challenging because it requires anticipating the model's systemic errors.

Consequently, we define the problem of blindspot discovery as the problem of finding an image classification model's systemic errors without making any of these assumptions. More formally, suppose that we have an image classifier, $f$, and a dataset of labeled images, $D = [x_i]_{i=1}^n$. Then, a *blindspot* is a coherent (*i.e.,* semantically meaningful) set of images, $\Psi \subset D$, where $f$ performs significantly worse (*i.e.,* $p(f, \Psi) \ll p(f, D \setminus \Psi)$ for some performance metric, $p$, such as recall). We denote the set of $f$'s *true blindspots* as $\mathbf{\Psi} : \{\Psi_m\}_{m=1}^M$. Next, we define the problem of *blindspot discovery* as the problem of finding $\mathbf{\Psi}$ using only $f$ and $D$. Then, a BDM is a method that takes as input $f$ and $D$ and outputs an ordered (by some definition of importance) list of *hypothesized blindspots*, $\hat{\mathbf{\Psi}} : [\hat{\Psi}_k]_{k=1}^K$. Note that the $\Psi_m$ and $\hat{\Psi}_k$ are sets of images.

**Approaches to BDM evaluation.** We observe that existing approaches to quantitatively evaluate BDMs fall in two categories. The first category of evaluations simply measure the error rate or size of $\hat{\Psi}_k$ (Singla et al., 2021; d'Eon et al., 2021). However, these evaluations have two problems. First, none of the properties they measure capture whether $\hat{\Psi}_k$ is coherent (*e.g.,* a random sample of misclassified images has high error but may not match a single semantically meaningful description). Second, $f$'s performance on $\hat{\Psi}_k$ may not be representative of $f$'s performance on similar images because BDMs are optimized to return high error images (*e.g.,* suppose that $f$ has a 90% accuracy on images of "squares and blue circles"; then, by returning the 10% of such images that are misclassified, a BDM could mislead us into believing that $f$ has a 0% accuracy on this type of image).

The second category of evaluations compares $\hat{\mathbf{\Psi}}$ to a subset of $\mathbf{\Psi}$ that have either been artificially induced or previously found (Sohoni et al., 2020; Eyuboglu et al., 2022). While these evaluations address the issues with those from the first category, they require knowledge of $\mathbf{\Psi}$, which is usually incomplete (*i.e.,* we usually only know a subset of $\mathbf{\Psi}$). This incompleteness makes it difficult to identify factors that influence BDM performance or to measure a BDM's recall or false positive rate. It is fundamentally impossible to fix this incompleteness using real data because we cannot enumerate

Table 1: A high level overview the high-level design choices made by different BDMs.

| Method | 1. Image Representation | 2. Dimensionality Reduction | 3. Hypothesis Class |
|---|---|---|---|
| Multiaccuracy (Kim et al., 2019) | VAE representation | | Linear model |
| GEORGE (Sohoni et al., 2020) | Model representation | UMAP ($d = 0, 1, 2$) | Gaussian kernels |
| Spotlight (d'Eon et al., 2021) | Model representation | | Gaussian kernels |
| Barlow (Singla et al., 2021) | Adversarially-Robust Model representation | | Decision Tree |
| Domino (Eyuboglu et al., 2022) | CLIP representation | PCA ($d = 128$) | Gaussian Kernels |
| `PlaneSpot` | Model representation | scvis ($d = 2$) | Gaussian Kernels |

all of the possible coherent subsets of $D$ to check if they are blindspots. To address these limitations, we introduce `SpotCheck`, which gives us complete knowledge of $\mathbf{\Psi}$ by using synthetic data.

**High-level design choices of BDMs.** In Table 1, we summarize three of the high-level design choices made by existing BDMs. First, each BDM uses a model to extract an *image representation*. Many BDMs use a representation from $f$, but some use pre-trained external models or other models trained on $D$. Second, some of the BDMs apply some form of *dimensionality reduction* to that image representation. Third, each BDM learns a model from a specified *hypothesis class* to predict if an image belongs to a blindspot from that image's (potentially reduced) representation.

Interestingly, while there has been significant effort focused on the choice of a BDM's image representation and hypothesis class (along with its associated learning algorithm), we note that dimensionality reduction has received much less attention. This is surprising because these BDMs are all solving clustering or learning problems, which are generally easier in lower dimensions. Motivated by this gap in prior work, we introduce a simple BDM, `PlaneSpot`, that uses a 2D representation.

## 3 EVALUATING BDMs USING KNOWLEDGE OF THE TRUE BLINDSPOTS

In Section 3.1, we introduce `SpotCheck`, which is an evaluation framework for BDMs that gives us complete knowledge of the model's true blindspots by using synthetic images and allows us to identify factors that influence BDM performance. In Section 3.2, we define the metrics that we use to measure the performance of BDMs given knowledge of the model's true blindspots.

### 3.1 SPOTCHECK: A SYNTHETIC EVALUATION FRAMEWORK FOR BDMs

`SpotCheck` builds on ideas from Kim et al. (2022) by generating synthetic datasets of varying complexity and training models to have specific blindspots on those datasets. We summarize its key steps below; see Appendix A for details.

**Dataset Definition.** Each dataset is defined using *semantic features* that describe the possible types of images it contains. Datasets that have a larger number of features have a larger variety of images and are therefore more complex. For example, a simple dataset may only contain images with squares and blue or orange circles (see Figure 1), while a more complicated dataset may also contain images with striped rectangles, small text, or grey backgrounds.

**Blindspot Definition.** Each blindspot is defined using a subset of the semantic features that define its associated dataset (see Figure 1). Similarly to how a dataset with more features is more complex, a blindspot defined using more semantic features is more specific.

**Training a Model to have Specific Blindspots.** For each dataset and blindspot specification, we train a ResNet-18 model (He et al., 2016) to predict whether a square is present. To induce blindspots, we generate data where the label for each image in the training and validation sets is correct if and only if it does not belong to any of the blindspots (see Figure 1). The test set images are always correctly labeled. Then, because we know the full set of semantic features that define the dataset, we can verify that the model learned to have exactly the set of blindspots that we intended it to. As a result, `SpotCheck` gives us complete knowledge of the model's true blindspots.

**Generating Diverse Experimental Configurations (ECs).** Since our goal is to study how various factors influence BDM performance, we generate a diverse set of *experimental configurations*, (*i.e.,* dataset, blindspots, and model triplets). To do this, we randomize several *factors*: the features that define a dataset (both the number of them and what they are) as well as the blindspots (the number

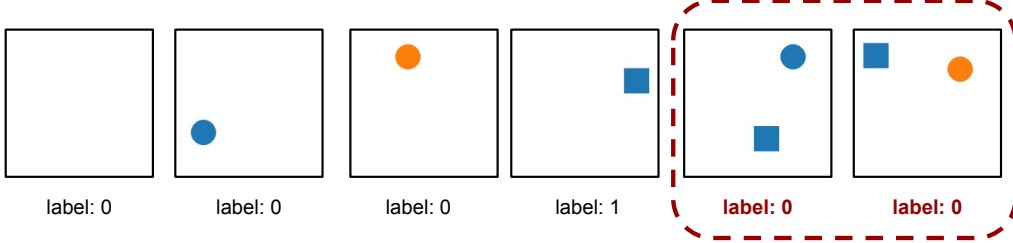

Figure 1: By controlling the data generation process, SpotCheck gives us complete knowledge of a model's true blindspots. Here, we show images sampled from a dataset created with SpotCheck. **Dataset Complexity.** This dataset is defined by 3 semantic features that vary across images: the presence of a square, the presence of a circle, and the color of the circle. We do not count the "color of the square" because it is always blue. **Blindspot Specificity.** This blindspot is defined by 2 semantic features: the presence of a square and the presence of a circle. As a result, it contains any image with both a square and a circle, regardless of the circle's color. **Data Labels.** In general, the label indicates if a square is present. However, training images belonging to this blindspot are mislabeled.

of them, the number of features that define them, and what those features are). Importantly, we sample these factors independently of one another across this set of ECs so that we can estimate their individual influence on BDM performance.

## 3.2 Evaluation Metrics based on Knowledge of the True Blindspots

These metrics measure how well the hypothesized blindspots returned by a BDM, $\hat{\mathbf{\Psi}}$, capture a model's true blindspots, $\mathbf{\Psi}$. Recall that the $\hat{\Psi}_k$ and $\Psi_m$ are sets of images. To do this, we start by measuring how well a BDM finds each *individual* true blindspot (Blindspot Recall) and build on that to measure how well a BDM finds the *complete set* of true blindspots (Discovery Rate and False Discovery Rate); see Appendix B for a discussion of how these metrics refine prior definitions.

**Blindspot Precision.** If $\hat{\Psi}_k$ is a subset of $\Psi_m$, we know that the model underperforms on $\hat{\Psi}_k$ and that $\hat{\Psi}_k$ is coherent. We measure this using the precision of $\hat{\Psi}_k$ with respect to $\Psi_m$:

$$\text{BP}(\hat{\Psi}_k, \Psi_m) = \frac{|\hat{\Psi}_k \cap \Psi_m|}{|\hat{\Psi}_k|} \tag{1}$$

Then, we say that $\hat{\Psi}_k$ *belongs to* $\Psi_m$ if, for some threshold $\lambda_p$:

$$\text{BP}(\hat{\Psi}_k, \Psi_m) \geq \lambda_p \tag{2}$$

However, $\hat{\Psi}_k$ can belong to $\Psi_m$ without capturing the same information as $\Psi_m$. For example, $\hat{\Psi}_k$ could be "squares and blue circles" while $\Psi_m$ could be "squares and blue or orange circles". Because this excessive specificity could result in the user arriving at conclusions that are too narrow, we need to incorporate some notion of recall into the evaluation.

**Blindspot Recall.** One way to incorporate recall is using the proportion of $\Psi_m$ that $\hat{\Psi}_k$ covers:

$$\text{BR}_{\text{naive}}(\hat{\Psi}_k, \Psi_m) = \frac{|\hat{\Psi}_k \cap \Psi_m|}{|\Psi_m|} \tag{3}$$

We relax this definition by allowing $\Psi_m$ to be covered by the union of the $\hat{\Psi}_k$ that belong to it:

$$\text{BR}(\hat{\mathbf{\Psi}}, \Psi_m) = \frac{\left|\left(\bigcup_{\hat{\Psi}_k : \text{BP}(\hat{\Psi}_k, \Psi_m) \geq \lambda_p} \hat{\Psi}_k\right) \cap \Psi_m\right|}{|\Psi_m|} \tag{4}$$

Then, we say that $\hat{\boldsymbol{\Psi}}$ *covers* $\Psi_m$ if, for some threshold $\lambda_r$:

$$\text{BR}(\hat{\boldsymbol{\Psi}}, \Psi_m) \geq \lambda_r \tag{5}$$

We do this because "squares and blue circles" and "squares and orange circles" belong to and jointly cover "squares and blue or orange circles." So, if a BDM returns both, a user could combine them to arrive at the correct conclusion.

**Discovery Rate (DR).** We define the DR of $\hat{\boldsymbol{\Psi}}$ and $\boldsymbol{\Psi}$ as the fraction of the $\Psi_m$ that $\hat{\boldsymbol{\Psi}}$ covers:

$$\text{DR}(\hat{\boldsymbol{\Psi}}, \boldsymbol{\Psi}) = \frac{1}{M} \sum_m \mathbb{1}(\text{BR}(\hat{\boldsymbol{\Psi}}, \Psi_m) \geq \lambda_r) \tag{6}$$

**False Discovery Rate (FDR).** When the DR is non-zero, we define FDR of $\hat{\boldsymbol{\Psi}}$ and $\boldsymbol{\Psi}$ as the fraction of the $\hat{\Psi}_k$ that do not belong to any of the $\boldsymbol{\Psi}$:[2]

$$\text{FDR}(\hat{\boldsymbol{\Psi}}, \boldsymbol{\Psi}) = \frac{1}{K} \sum_k \mathbb{1}(\max_m BP(\hat{\Psi}_k, \Psi_m) < \lambda_p) \tag{7}$$

Note that it is impossible to calculate the FDR without the complete set of true blindspots. While `SpotCheck` gives us this knowledge, it is generally not available.

## 4 PLANESPOT: A SIMPLE BDM BASED ON DIMENSIONALITY REDUCTION

In this section, we define `PlaneSpot`. As shown in Table 1, `PlaneSpot` uses the most common choices for the image representation (*i.e.,* $f$'s own representation) and the hypothesis class (*i.e.,* gaussian kernels). `PlaneSpot` also uses standard techniques to learn a model from that hypothesis class. As a result, the most interesting aspect of `PlaneSpot`'s design is that it finds blindspots using a 2D representation. We start by defining some additional notation and then explain `PlaneSpot`'s choice for each of the high-level design choices made by a BDM.

**Notation.** Suppose that we want to find $f$'s blindspots for a class, $c$, and let $D^c$ be the set of images from $D$ that belong to $c$. Further, suppose that we have divided $f$ into two parts: $g : X \to \mathbb{R}^d$, which extracts $f$'s representation of an image (*i.e.,* its penultimate layer activations), and $h : \mathbb{R}^d \to [0, 1]$, which gives $f$'s predicted confidence for $c$.

**Image Representation.** We use $g$ to extract $f$'s representation for $D^c$, $G = g(D^c) \in \mathbb{R}^{n \times d}$, and $h$ to extract $f$'s predicted confidences for $D^c$, $H = h(G) \in [0, 1]^{n \times 1}$. Note that, because all of the images in $D^c$ belong to $c$, entries of $H$ closer to 1 denote higher confidence in the true class.

**Dimensionality Reduction.** We use $G$ to train scvis (Ding et al., 2018), which combines the objective functions of tSNE and an autoencoder, in order to learn a 2D representation of $f$'s representation, $s : \mathbb{R}^d \to \mathbb{R}^2$. Then, we use $s$ to get the 2D representation of $D^c$, $S = s(G) \in \mathbb{R}^{n \times 2}$. Finally, we normalize the columns of $S$ to be in $[0, 1]$, $\bar{S} \in [0, 1]^{n \times 2}$.

**Hypothesis Class.** We want `PlaneSpot` to be aware of both the representation of $D^c$, $\bar{S}$, and $f$'s predicted confidences for $D^c$, $H$, so we combine them into a single representation, $R = [\bar{S}; w \cdot H] \in \mathbb{R}^{n \times 3}$ where $w$ is a hyperparameter controlling the relative weight of these components. Then, we pass $R$ to a Gaussian Mixture Model clustering algorithm, where the number of clusters is chosen using the Bayesian Information Criteria. Finally, we return those clusters in order of decreasing importance, which we define using the product of their error rate and the number of errors in them.

## 5 EXPERIMENTS

In Section 5.1, we use `SpotCheck` to run a series of controlled experiments using synthetic data. In Section 5.2, we demonstrate how to setup semi-controlled experiments to validate that the findings from `SpotCheck` generalize to settings with real data.

---

[2]While calculating DR, we may only need the top-$u$ items of $\hat{\boldsymbol{\Psi}}$. As a result, we only calculate the FDR over those top-$u$ items. This prevents the FDR from being overly pessimistic when we intentionally pick $K$ too large in our experiments. However, it is unclear what value of $u$ to use for ECs that have a DR of zero, so we exclude these ECs from our FDR analysis.

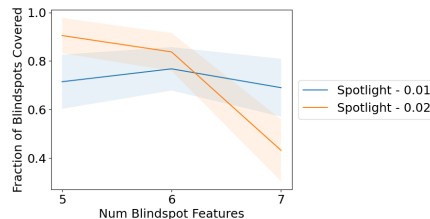

Table 2: Average BDM DR and FDR with their standard errors across 100 `SpotCheck` ECs.

| Method | DR | FDR |
|---|---|---|
| Barlow | 0.43 (0.04) | 0.03 (0.01) |
| Spotlight | 0.79 (0.03) | 0.09 (0.01) |
| Domino | 0.64 (0.04) | 0.07 (0.01) |
| **PlaneSpot** | **0.88 (0.03)** | **0.02 (0.01)** |

Figure 2: The fraction of blindspots covered (with shaded 95% CIs) for blindspots defined using 5, 6, and 7 features using 2 different Spotlight hyperparameter choices.

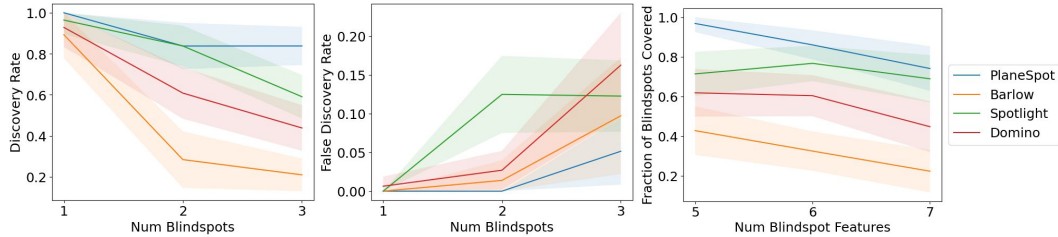

Figure 3: (Left/Center) Average BDM DR/FDR (with shaded 95% confidence intervals) for ECs that have 1, 2, and 3 blindspots. (Right) The fraction of blindspots covered, across the individual blindspots from the ECs, for blindspots defined using 5, 6, and 7 features.

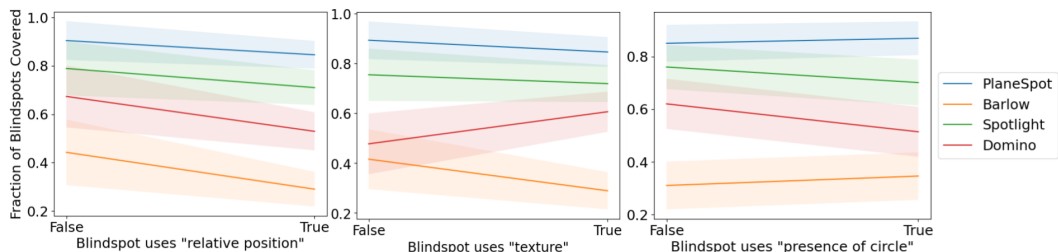

Figure 4: The fraction of blindspots covered (with shaded 95% confidence intervals) that are or are not defined with (Left) the "relative position" feature (whether the square appears above the centerline of the image), (Center) a "texture" feature (whether any object in the image has vertical stripes), or (Right) the presence of a circle (whether there is a circle in the image).

## 5.1 SYNTHETIC DATA EXPERIMENTS

We use `SpotCheck` to generate 100 ECs whose datasets have 6-8 semantic features and whose models have 1-3 blindspots defined with 5-7 features. We evaluate three recent BDMs: Spotlight (d'Eon et al., 2021), Barlow (Singla et al., 2021), Domino (Eyuboglu et al., 2022), and `PlaneSpot`. We give all BDMs the positive examples (*i.e.,* images with squares) from the test set and limit them to returning 10 hypothesized blindspots. We use a held-out set of 20 ECs to tune each BDM's hyperparameters (see Appendix C). We use $\lambda_p = \lambda_r = 0.8$ for our metrics.

**Overall Results.** Table 2 shows the DR and FDR results averaged across all 100 ECs. We observe that `PlaneSpot` has the highest DR and that `PlaneSpot` and Barlow have a lower FDR than Spotlight and Domino. In Appendix D, we take a deeper look at why these BDMs are failing and conclude that a significant portion of all of their failures can be explained by their tendency to merge multiple true blindspots into a single hypothesized blindspot.

**Identifying factors that influence BDM performance.** We study two types of factors: *per-configuration factors*, which measure properties of the dataset (*e.g.,* how complex is it?) or of the model (*e.g.,* how many blindspots does it have?), and *per-blindspot factors*, which measure properties of a blindspot (*e.g.,* is it defined with this feature?). For per-configuration factors, we average DR and FDR across the ECs. For per-blindspot factors, we report the fraction of blindspots covered averaged across each individual blindspot from the ECs (see Equation 5).

**The number of blindspots matters.** In Figure 3 (Left), we plot the average DR for ECs with 1, 2, and 3 blindspots. Average DR decreases for all methods as the number of blindspots increases. Figure 3 (Center) shows that FDR increases as the number of blindspots increases. Together these observations show that BDMs perform worse in settings with multiple blindspots, which is particularly significant because past evaluations have primarily focused on settings with one blindspot.

**The specificity of blindspots matters.** In Figure 3 (Right), we plot the fraction of blindspots covered for blindspots defined using 5, 6, and 7 features. With the exception of Spotlight, all of these methods are less capable of finding more specific/less frequently occurring blindspots.

**The features that define a blindspot matter.** In Figure 4, we plot the fraction of blindspots covered for blindspots that either are or are not defined using various features. In general, we observe that the performance of these BDMs is influenced by the types of features used to define a blindspot (*e.g.,* the presence of spurious objects, color or texture information, background information). For example, all methods are less likely to find blindspots defined using the "relative position" feature. Interestingly, BDMs that use the model's representation for their image representation (*i.e.,* `PlaneSpot` and Spotlight) are less sensitive to the features that define a blindspot than those that use an external model's representation (*i.e.,* Barlow and Domino).

**Hyperparameters matter.** We make two observations about BDM hyperparameters that, jointly, suggest that it is critical that future BDMs provide ways to tune their hyperparameters. First, in Appendix C, we observe that many BDMs have significant performance differences when we vary their hyperparameters. However, hyperparameter tuning is harder in real settings than in these experiments because there is no information about the true blindspots to use. Second, in Figure 2, we observe that two hyperparameter settings that perform nearly identically on average exhibit significantly different performance at finding blindspots defined using differing numbers of features. This suggests that there may not be a single best hyperparameter choice to find all of the blindspots in a single model, which could contain multiple blindspots of different specificity or size.

## 5.2 REAL DATA EXPERIMENTS

We demonstrate how to design semi-controlled experiments using the COCO dataset (Lin et al., 2014) to test whether or not some of the findings observed using `SpotCheck` generalize to settings with real data. Despite the fact COCO is a fairly large dataset with extensive annotations, it does not have enough metadata to test all of the findings from `SpotCheck` (*e.g.,* we cannot induce blindspots that depend on the texture of an object). Therefore, we study the following questions using real data:

- Are BDMs still less effective for models with more true blindspots?
- Are BDMs still less effective at finding more specific blindspots?
- Will the 2D image representation used by `PlaneSpot` still be effective?

Notice that means we we are interested in two factors: the *number* of blindspots in a model and of the *specificity* of those blindspots. Then, in order to try to estimate their influence on BDM performance, we use the same strategy as `SpotCheck` and generate a set of ECs where we randomize these factors independently of one another. We provide details of how we generate these ECs in Appendix E and show an example EC in Figure 5.

However, one key difference from our synthetic experiments is that when we use real data, we cannot guarantee that the model will learn the blindspots that we try to induce in it. Then, if the model fails to learn the intended blindspots in a non-random way, the factors that we want to investigate may be correlated and, consequently, their effects may be confounded, which makes it difficult to estimate their individual effects. Table 3 shows that this confounding occurs between the effects of the number and the specificity of the blindspots in the COCO ECs. To eliminate this confounding, we generate a new set of ECs for each factor where we hold the other confounding factor constant.

**blindspot 1: elephants with people**

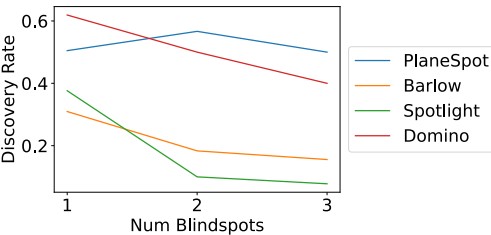

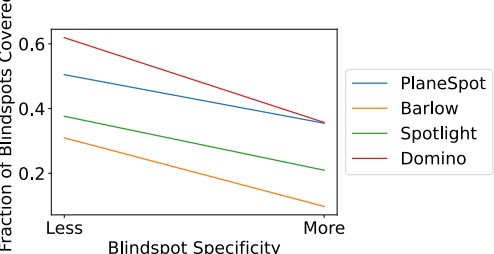

label: 0    label: 1    label: 0    **label: 0**    **label: 0**    **label: 0**

**blindspot 2: zebras**

Figure 5: Images sampled from an example EC created using COCO. In this example, the task is predict to detect whether an animal is present. The EC has two true blindspots: "elephants with people," which is more-specific because it is defined using two objects, and "zebras," which is less-specific because it is defined using only one object.

Table 3: The effect of number of blindspots is confounded with the effect of the specificity of those blindspots: as we try to induce more blindspots in the model, the model is less likely to learn more specific blindspots. Overall, the model learned the intended blindspots in 65 of the 90 ECs in the general pool.

Table 4: For the 65 ECs where the model learned the intended blindspots, we report BDM DR with its standard error. We only have knowledge of the blindspots that we induced, so the DR is calculated relative to those blindspots and we cannot calculate the FDR.

| Num Blindspots | Fraction of successfully induced Blindspots that are more specific |
|---|---|
| 1 | 0.54 |
| 2 | 0.47 |
| 3 | 0.40 |

| Method | DR |
|---|---|
| Barlow | 0.09 (0.03) |
| Spotlight | 0.14 (0.04) |
| Domino | 0.38 (0.05) |
| **PlaneSpot** | **0.48** (0.05) |

Figure 6: When we condition on only having less specific blindspots, we see that BDM performance decreases as we increase the number of blindspots. These results are based on 82 ECs where the model learned the intended blindspots.

Figure 7: When we condition on only having a single blindspot, we see that BDMs are less effective at finding more specific blindspots. These results are based on 70 ECs where the model learned the intended blindspots.

We call the set of ECs, where all of the factors are chosen independently, the *general pool* because it covers a wider range of possible ECs, which is better for measuring overall BDM performance. We call this new set of ECs the *conditioned pool* because it is generated by conditioning the distribution used to generate the general pool on specific confounding factors. Because blindspot discovery is harder with real data, we use more lenient thresholds for our metrics and set $\lambda_p = \lambda_r = 0.5$.

**Results.** Overall, all of the findings that we observed using `SpotCheck` generalized to these real data experiments. In Figure 6, using the conditioned pool, we observe that BDM performance generally decreases as the number of blindspots increases. In Figure 7, using the conditioned pool, we observe that these BDMs are less able to find blindspots that are more specific. In Table 4,

using the general pool, we observe that `PlaneSpot` is still effective on real data. These results provide evidence that results from `SpotCheck` may generalize to a wider range of more realistic settings. Additionally, in Appendix F, we include qualitative examples of the 2D representation used by `PlaneSpot` for models trained on benchmark datasets.

**Interpreting these results.** While these results are promising, there are general challenges associated with running semi-controlled experiments on real data that influence the interpretation of these results (or the results of any similar experiments). At a high-level, these challenges all add noise (or bias) to the results.

First, models trained on real data probably have naturally occurring blindspots (*i.e.,* blindspots that we did not intend the model to learn) that may influence the results. One example of how they do this is that we do not know how many hypothesized blindspots to ask a BDM to return because we do not know how many true blindspots (including both the naturally occurring ones and the ones we induced) a model has.

Second, the chosen BDM hyperparameters are probably sub-optimal because we cannot tune them for a specific EC. Because the set of ECs from the synthetic data experiments is relatively homogeneous, it is reasonable to use a single set of hyperparameters for all of those ECs. However, this is not the case for our COCO experiments.

Third, the results may be biased in potentially unknown ways towards certain BDMs whenever some of the models in the set of ECs fail to learn the intended blindspots. For example, our general pool of ECs contains more less-specific blindspots than more-specific blindspots, which will bias the results towards BDMs that are more effective at finding less-specific blindspots. Further, there are probably other biases that we cannot identify because we lack the metadata needed to measure them.

Fourth, there can be false positives in verifying that a model actually learned an intended blindspot. This is because there could be a sub-blindspot (that we may not have the metadata to define) within that intended blindspot that is actually causing the model to perform significantly worse on the intended blindspot.

## 6 CONCLUSION

In this work, we introduced `SpotCheck`, a synthetic evaluation framework for BDMs, and `PlaneSpot`, a simple BDM that uses a 2D image representation. Using `SpotCheck`, we identified factors that influence BDM performance and two technical challenges for future work on developing new BDMs to address (*i.e.,* providing a practical way to tune BDM hyperparameters and preventing BDMs from merging multiple true blindspots into a single hypothesized blindspot). We found that `PlaneSpot` outperforms existing BDMs on both synthetic and real data.

Beyond the insights made in the experiments that we ran, we believe that our demonstrated evaluation workflow has laid further groundwork for a more rigorous science of blindspot discovery. Researchers interested in benchmarking their own BDM or answering scientific questions about blindspot discovery can first use `SpotCheck` (or harder variations of it) to run controlled experiments using synthetic data. Next, researchers can also set up semi-controlled experiments using real data. While both harder and noisier (in part, because of the challenges that we identified), these experiments are also more realistic. A natural open direction for future work is how to evaluate BDMs in more realistic settings where we have no knowledge of the model's true blindspots.

We note that this workflow is particularly well suited to identifying factors that influence BDM performance and that understanding these factors can benefit both researchers and practitioners using BDMs. For researchers, understanding these factors is essential to running fair and informative experiments (*e.g.,* only evaluating BDMs in settings that only have one blindspot, which we have identified as being easier, may misrepresent how BDMs perform in more realistic settings). For practitioners, understanding these factors may help them use any prior knowledge they may have to select an effective BDM for their problem. Two potential factors that future work could consider are: First, how does the methodology used to induce blindspots (*i.e.,* training with mislabeled images) influence BDM performance? Second, for models trained on real data, how different are the naturally occurring blindspots from those that we have the metadata to artificially induce and how do those differences influence BDM performance?

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
