# OpenReview forum: "Towards a More Rigorous Science of Blindspot Discovery in Image Models"
_ICLR.cc/2023/Conference — Submitted to ICLR 2023_

### Official Review · Reviewer_X8MU · 2022-10-23

**Confidence:** 4
**Correctness:** 2
**Technical Novelty And Significance:** 1
**Empirical Novelty And Significance:** 1
**Recommendation:** 3

**Clarity, Quality, Novelty And Reproducibility:**

The writing of the paper could be made clearer, especially details on the synthetic ECs. My concerns with novelty/usefulness of findings are discussed above.

**Strength And Weaknesses:**

Blindspot detection is an important problem and understanding how to quantify the effectiveness of different approaches and identify the factors that influence them in real datasets could be valuable.

However, there seems to be a huge gap between the simplistic settings the authors study in this paper and real-world datasets. That would still be ok if the authors showed how their analysis could help improve blindspot detection in more realistic settings (eg, on ImageNet). As of now, the paper just reads as an analysis of a simple setting that may or may not be reflective of practice. As the authors themselves mention in the paper, it is not clear whether model/BDM trends on artificially induced blindspots are predictive of trends on naturally occurring ones.

More specific comments/questions:

- In general, in the results in Figures 3, 4 and 6, one thing that is not clear is how sensitive the model is actually to these blindspots. In particular, an important quantity to visualize is model performance on blindspot datasets as the number of blindspots/specificity, etc. is varied. After all, just because the dataset has a blindspot doesn’t mean the model learns it. It is entirely possible that as the number of blindspots increase, the model tend to not learn all of them. This is a key confounder in the results that must be ablated.
- What exactly are the features in the synthetic dataset?

**Summary Of The Paper:**

This paper puts forth a methodology to understand the underlying factors (in the model and/or data) that affect blindspot detection methods. They rely on synthetic data wherein the number and nature of blindspots can be controlled. They also propose a new blindspot detection approach, PlaneSpot.

**Summary Of The Review:**

Overall, the analysis in the paper and its connections with blindspots in practical models on large-scale datasets is not yet sufficiently compelling. Therefore, I recommend rejection.

---

> ### Author Response · Authors · 2022-12-12
> **Response**
>
>
> Thank you for the review.  We answer your questions and clarify several points below.
>
> **Snippet**: "there seems to be a huge gap between the simplistic settings the authors study in this paper and real-world datasets. That would still be ok if the authors showed how their analysis could help improve blindspot detection in more realistic settings (eg, on ImageNet)"
> - See General Response Point #1.
>
> **Snippet**: "just because the dataset has a blindspot doesn’t mean the model learns it. It is entirely possible that as the number of blindspots increase, the model tend to not learn all of them. This is a key confounder in the results that must be ablated"
> - For both our synthetic and real (COCO) data experiments, we clarify that **we do validate that the models we use in our evaluations do learn all of the true blindspots**.  We use a separate hold-out set to validate that the model has worse performance (at least 20% worse recall) on images belonging to the blindspot.
> -  We exclude all models that did not learn all of the true blindspots from further analysis.  For the synthetic experiments, we successfully induced all of the true blindspots.  For the COCO experiments, We include details about the % of models we trained where we were able to successfully learn all of the true blindspots in Appendices E.3-E.4.
> -  We design our experiments to control for the confounding effect noted by the reviewer that comes from models failing to learn the true blindspots.  We explain how we address confounding in Section 5.2 (with further details in Appendices E.3-E.4).
>
> **Snippet**: "in the results in Figures 3, 4 and 6, one thing that is not clear is how sensitive the model is actually to these blindspots [...] an important quantity to visualize is model performance on blindspot datasets as the number of blindspots/specificity, etc. is varied"
> - For the synthetic data results reported in Figures 3 and 4, all of the models that were used had near-zero accuracy for images in each blindspot, and near-perfect accuracy on images that did not belong to a blindspot (see General Response Point #2).
> - For the real COCO data results reported in Table 4 and Figures 6 and 7,  the model had 20-80% lower recall on the blindspots.   We observe that all BDMs find a much larger % of blindspots that have larger accuracy gaps.
>
> **Snippet**: "What exactly are the features in the synthetic dataset?"
> - The semantic features used to generate synthetic images are listed in Table 5 in Appendix A.1, and include the presence, size, color, texture, number, or relative location of objects present in each image.
>
> **Snippet**: "the analysis in the paper and its connections with blindspots in practical models on large-scale datasets is not yet sufficiently compelling"
> -  We respectfully disagree with the claim that we did not evaluate on "large-scale datasets": we evaluate recent state-of-the-art BDMs using a subset of Microsoft Common Objects in Context (COCO), a dataset containing over 100,000 photographs [1].
>
> **References:**
>
> [1] Tsung-Yi Lin, Michael Maire, Serge Belongie, James Hays, Pietro Perona, Deva Ramanan, Piotr Dolla ́r, and C Lawrence Zitnick. Microsoft coco: Common objects in context. In European conference on computer vision, pp. 740–755. Springer, 2014.

---

### Official Review · Reviewer_E6Ce · 2022-10-24

**Confidence:** 4
**Clarity, Quality, Novelty And Reproducibility:** See above
**Correctness:** 3
**Technical Novelty And Significance:** 2
**Empirical Novelty And Significance:** 2
**Recommendation:** 3

**Strength And Weaknesses:**

The paper tackles an import problem: providing a taxonomy and evaluation framework for these "blind spot detection" methods. However, the work suffers from a number of issues around the completeness of the real world experiments and clarity.

Real world experiments: All of the experiments are synthetic. Even on the real world experiments, the authors only evaluated on synthetic blindspots that are not representative of conditions in the real world. While this is an interesting experiment, the authors must go beyond these synthetic experiments to identify naturally arising blindspots in models --- otherwise it is not clear whether the presented method is actually useful. The approaches to doing so are relatively straightforward: the authors could
* evaluate if identified blindspots match known blindspots of examples (for example, on ImageNet humans can lead to fish), or
* investigate identified blindspots (test whether or not these blindspots actually matter with counterfactuals or another.

Clarity:
* Missing definition of semantic meaningfulness: what makes a blindspot semantically meaningful? This is a critical question that defines the conceptual objects investigated in this paper. What is to stop one from outputting all the incorrect examples as a subpopulation? After all, incorrect examples are semantically meaningful as a unit in that they are all incorrect.
* Notation: The paper uses a large amount of notation. There are many symbols, acronyms, and named methods that clutter the messages that the paper tries to get across. For example, Table 2.
* Confidence intervals: Not defined; are they bootstrapped?
* Figure 7: What does the x axis mean here? This specificity should be a number value.

**Summary Of The Paper:**

 In this work the authors give a framework for evaluating blindspot detection methods (BDMs). The authors define BDMs as methods taking labeled examples and a classifier, and outputting "semantically meaningful" groups of points. The authors introduce a new BDM, and a toy evaluation setting in which they perform image classification under a known distribution of images and a known set of "blind spots." The introduced method, PlaneSpot, outperforms the other compared-with methods in both this toy setting and in a more realistic setting.

**Summary Of The Review:**

The paper studies an important problem, but does not have experimentally convincing results.

---

> ### Author Response · Authors · 2022-12-12
> **Response**
>
> Thank you for the review.  We are glad you agree that our work "tackles an important problem: providing a taxonomy and evaluation framework" for blindspot discovery.  We respond to your questions and comments below.
>
> **Snippet**: "While this is an interesting experiment, the authors must go beyond these synthetic experiments to identify naturally arising blindspots in models --- otherwise it is not clear whether the presented method is actually useful"
> - Please see General Response Points #1 and #2.
>
> **Snippet**: "what makes a blindspot semantically meaningful? [...] What is to stop one from outputting all the incorrect examples as a subpopulation?"
> - Our work defines a blindspot as a set of input images $\Psi \subset D := [x_i]_{i=1}^n$.  Like prior work, we define "semantically meaningful" (i.e. "coherent") as "united by a human-understandable concept" [1].
> - We note that while coherence is subjective and difficult to define formally, a reasonable litmus test of group coherence is whether all of the images in the group can be described (i.e. "images that have both elephants and people") in a way such that other people can easily determine whether a new input $x'$ belongs to the group.
> - While the reviewer's example of a group that contains "all incorrect examples" can be described, it may not meet the above definition of coherence, as absent information about the model's prediction a user cannot easily determine whether a new image $x'$ matches the text description.
>
> **Snippet**: "Confidence intervals: Not defined; are they bootstrapped?"
> - To calculate the CIs, we first calculate the empirical mean and standard deviation of the metric across all relevant experimental configurations.  We define all 95% CIs as the empirical mean plus or minus 1.96 standard deviations.
>
> **Snippet**: "Figure 7: What does the x axis mean here? This specificity should be a number value"
> - For the COCO results shown in Figure 7, "less specific" blindspots are defined using 1 COCO object category (e.g. "elephants"), and "more specific" blindspots are defined using 2 COCO object categories (e.g. "elephants with people").  We provide further details in Appendix E.2.
> - In the current paper draft, we implicitly define "more" and "less" specific in the caption explaining the example provided in Figure 5.  We will update the manuscript for clarity.
>
>
> **References**
>
> [1] Sabri Eyuboglu, Maya Varma, Khaled Kamal Saab, Jean-Benoit Delbrouck, Christopher Lee-Messer, Jared Dunnmon, James Zou, and Christopher Re. Domino: Discovering systematic errors with cross-modal embeddings. In International Conference on Learning Representations, 2022. URLhttps://openreview.net/forum?id=FPCMqjI0jXN.

---

### Official Review · Reviewer_w8rw · 2022-10-26

**Confidence:** 3
**Correctness:** 3
**Technical Novelty And Significance:** 3
**Empirical Novelty And Significance:** 3
**Recommendation:** 6

**Clarity, Quality, Novelty And Reproducibility:**

The paper is very clear overall, but some further clarifications related to the claims is required as explained in the previous section. The quality of the work is high. The work is also relatively novel in that it tries to formalize an existing but previously unformalized research area, and also provides a new benchmark and model for it. The work seems quite reproducible from what is written in the paper, except for Spotcheck that is only defined in terms of the high level concepts and not the specifics of the dataset itself. More specific details are necessary in the main paper for proper reproducibility.

**Strength And Weaknesses:**

Strengths:

Clear and concise writing.
The work attempts to provide a more clear formalization of Blindspot Discovery problem and they do so well.
The authors motivate the need for a more ‘controllable’ benchmark and provide one in a satisfactory way.
Good empirical evaluation of the new proposed model.

Weaknesses:

While the proposed benchmark is a good step forward, I do feel that the paper suffers from some ‘overclaiming’, in the sense that, even their proposed benchmark may have ‘naturally’ occurring blindspots. For example, the model itself may have blindspots when the data presents more than two objects, or perhaps when it presents two objects that are on the right hand side of the square, or on the top left corner of the square. These are blindspots that do not exist within the authors’ predefined schema for what label 0 and 1 belong to, but the model may none-the-less develop sensitivities to them. Frankly I think, trying to create a ‘perfect’ synthetic dataset for blindspot discovery can be a good way to create a blindspot for a research project. In the sense that perhaps a better way to frame the work would be an attempt to create a synthetic dataset with more specific and controllable attributes. However, and this leads to the second weakness.
It’s not clear what the difference between existing benchmarks on real world data, that have previously discovered or defined blindspots, and the proposed synthetic dataset are, since even the synthetic dataset can’t be said to not have ‘naturally’ occurring blindspots that the authors can’t predict. A better way to quantify or justify this is necessary I believe.
It’s not clear what the exact usefulness of the new benchmark is. Is it cheaper to evaluate on? Or is it the fact that it has more ‘controllable’ blindspots? If so, the above concerns must be addressed. Also, given the correlation between the real world datasets and the results from your synthetic dataset, wouldn’t you say that perhaps one could omit using SpotCheck and simply use the real world datasets?


**Summary Of The Paper:**

The authors first propose a formalization of the Blindspot Discovery problem, which they claim was not previously done in a clear manner, and then proceed to propose a synthetic image benchmark for evaluating blindspot discovery methods, that they call SpotCheck. Furthermore they propose a novel blindspot discovery method, called PlaneSpot which outperforms existing methods. Finally, the authors evaluate existing methods as well as their proposed method on real-data sources and find that the results of their proposed synthetic benchmark SpotCheck, seem to correlate with the results of the benchmarks sourced from real datasets. They therefore conclude that their benchmark can be a good complementary and thorough way of evaluating BDMs.

**Summary Of The Review:**

While I am expecting the authors to provide me with more information wrt to the weaknesses I outlined above, I also believe that the work represents a good step forward for BDMs, both in terms of better formalization and benchmarks as well as models.

---

> ### Author Response · Authors · 2022-12-12
> **Response**
>
> Thank you for your review.  We are excited that the reviewer believes our work is a "good step forward for BDMs" that provides "better formalization and benchmarks as well as models".  We respond to your comments below:
>
> **Snippet #1**: "I do feel that the paper suffers from some ‘overclaiming’, in the sense that, even their proposed benchmark may have ‘naturally’ occurring blindspots"
> - In the General Response Point #2, we clarify that models that we trained using **synthetic data** from SpotCheck **do not have "naturally occurring" blindspots**.
> - In our work, we state that the possibility of "naturally occurring blindspots" is one limitation of our real (COCO) data experiments.  We note that to our knowledge, prior controlled evaluations of BDMs such as [1] also have this limitation.
>
> **Snippet #2**: "the model itself may have blindspots when the data presents more than two objects, or perhaps when it presents two objects that are on the right hand side of the square, or on the top left corner of the square. These are blindspots that do not exist within the authors’ predefined schema"
> - We agree with the reviewer that there do exist other ways to define coherent subgroups of the synthetic data (such as the two examples given) that we do not explicitly include within our taxonomy of possible true blindspots (detailed in Appendix A.3).  We acknowledge that our space of possible blindspot definitions is not comprehensive, and that future extensions can be explored.
> - We selected the blindspot definitions used in our experiments because of their similarity to real blindspots discovered by past work, which were defined by the presence of spurious objects, or the object's color, texture, or relative position.
>
> **Snippet #3**: "It’s not clear what the difference between existing benchmarks [...] and the proposed synthetic dataset are, since even the synthetic dataset can’t be said to not have ‘naturally’ occurring blindspots that the authors can’t predict"
> - As stated in General Response Point #2, the models trained on synthetic data do not have "blindspots that the authors can't predict".
>
> **Snippet #4**:  "It’s not clear what the exact usefulness of the new benchmark is [...] given the correlation between the real world datasets and the results from your synthetic dataset, wouldn’t you say that perhaps one could omit using SpotCheck and simply use the real world datasets?"
> - We describe several benefits to conducting fully synthetic evaluations in General Response #2.
> - One limitation of using real data is that we observed that we would often fail to induce the intended blindspots, introducing confounding that posed difficulties when trying to measure the effects of different experimental factors.  To address this limitation, we had to design different pools of ECs to measure the effect of each experimental factor.  This is much more expensive than our synthetic experiments, which could use a single pool of ECs (as the models successfully underperformed on all of the induced blindspots).
>
> **Snippet #5**: "The work seems quite reproducible from what is written in the paper, except for Spotcheck that is only defined in terms of the high level concepts and not the specifics of the dataset itself. More specific details are necessary in the main paper"
> - We detail how to reproduce the process we used to generate the synthetic images used in our experiments in Appendix A.
>
> [1] Sabri Eyuboglu, Maya Varma, Khaled Kamal Saab, Jean-Benoit Delbrouck, Christopher Lee-Messer, Jared Dunnmon, James Zou, and Christopher Re. Domino: Discovering systematic errors with cross-modal embeddings. In International Conference on Learning Representations, 2022. URLhttps://openreview.net/forum?id=FPCMqjI0jXN.

---

### Official Review · Reviewer_pnqW · 2022-10-26

**Confidence:** 3
**Correctness:** 3
**Technical Novelty And Significance:** 2
**Empirical Novelty And Significance:** 3
**Recommendation:** 5

**Clarity, Quality, Novelty And Reproducibility:**

Clarity and Quality
- The weaknesses above are all about clarity. Thus, there is room for improvement.

Novelty
- The proposed framework or BDM is simple and has no major technical novelties. However, the reviewer thinks significant technical novelty is optional for such a benchmark and baseline proposal.

Reproducibility
- There is also a detailed description in the appendix, and the code is provided. Reproducibility is sufficient.

**Strength And Weaknesses:**

Strengths
1. The definition of BDM and SpotCheck is clear.
1. A simple BDM, PlaneSpot, is proposed using dimensionality reduction and Gaussian Mixture Model.
1. Experimental results on both synthesized and real datasets.

Weaknesses
1. Motivation to use scvis (Ding et al., 2018) as a dimensionality reduction method is unclear. There are many methods, such as PCA, LLE, t-SNE, and UMAP. The reviewer would like to know why scvis is selected.
1. It is also unclear how Ψ is generated using PlaneSpot. It seems clusters found by Gaussian Mixture Model are used for Ψ, but its particular definition is not described.
1. While the reviewer can agree that PlainSpot has the best accuracy on synthetic data, Domino sometimes outperforms PlainSpot on real data. This result is not merely a slight concern to say that PlainSpot is the best method among the other methods but also suggests that the evaluation of BDM by synthetic data proposed as SpotCheck may not correlate very strongly with the result using real data. In fact, there is also a discrepancy about the superiority of the methods, including Barlow and Spotlight, between the artificial data and the actual data.

**Summary Of The Paper:**

This paper addresses the issue of Blindspot Discovery Methods (BDM), which is the task of finding semantically meaningful subsets of data that significantly degrade the performance of image classifiers. This paper proposes SpotCheck as an evaluation framework using synthetic data and PlaneSpot as a new method. Experiments are also conducted on real data sets.

**Summary Of The Review:**

Overall, the reviewer leans toward rejecting this paper. A response from the authors regarding the above weaknesses could improve the score.

---

> ### Author Response · Authors · 2022-12-12
> **Response**
>
> Thank you for your review and for acknowledging strengths of the work.  We respond to your questions below:
>
> **Snippet:** "There are many [dimensionality reduction] methods, such as PCA, LLE, t-SNE, and UMAP.  The reviewer would like to know why scvis is selected".
> - We agree with the reviewer that dimensionality reduction is an underexplored aspect of BDM design, and that using a different dimensionality reduction method may lead to improved performance over SCVIS.  We note that our proposed framework can be easily used to evaluate new BDMs that use alternative dimensionality reduction methods.
> - We chose SCVIS because we believe there are similarities between SCVIS's initial use (identifying cell types) and blindspot discovery: both tasks require finding potentially small groups of points in a high dimensional space.
>
> **Snippet**: "It is also unclear how Ψ is generated using PlaneSpot. It seems clusters found by Gaussian Mixture Model are used for Ψ"
> - The reviewer is correct.  We will update the manuscript for clarity.
>
> **Snippet**: "While the reviewer can agree that PlainSpot has the best accuracy on synthetic data, Domino sometimes outperforms PlainSpot on real data. This result is not merely a slight concern to say that PlainSpot is the best method among the other methods but also suggests that the evaluation of BDM by synthetic data proposed as SpotCheck may not correlate very strongly with the result using real data. In fact, there is also a discrepancy about the superiority of the methods, including Barlow and Spotlight, between the artificial data and the actual data."
>
> -  See General Response Point #3, where we further explain the differences between Real vs. Synthetic data results.

---

### Author Response · Authors · 2022-12-12
**General Response**

We would like to thank the reviewers for their feedback and suggestions to improve our work.  We respond to related questions and comments in our General Response.

**Motivation.**  In our work, we design an extensible evaluation framework to achieve two goals:
- G1: Efficiently & rigorously evaluate a BDM
- G2: Help us understand what “experimental factors” influence BDM performance

We refer back to these 2 goals below to further motivate our experiments.

**Point #1: Why don't we evaluate using only "naturally occurring" blindspots? (Reviewers E6C3, X8MU)**
Reviewers E6Ce and X8MU asked why we did not compare the hypothesized blindspots returned by BDMs to "naturally occurring" blindspots.  We see two options for how to do this:
- Option 1:  Compare to naturally occurring blindspots documented in past work (e.g. for models trained on ImageNet).
  - Problem:  There are relatively few blindspots documented in past work.  Only comparing against past blindspots may encourage overfitting to well-documented examples and lead to misrepresentative conclusions.
- Option 2:  Run a user study, where we use humans to directly measure the "coherence" of the hypothesized blindspots.
  - Problem:  User studies are slow and difficult to run.

Neither of these options are well suited for either Goal 1 or Goal 2.  Therefore we "artificially induce" known blindspots so that we can efficiently compare the hypothesized blindspots to the induced model blindspots.

**Point #2: Usefulness of Evaluations with Synthetic Data (Reviewers w8rw, X8MU)**

Using synthetic data allows us to do three things:
- Easily manipulate different “experimental factors” that we want to study
- Ensure that the model always learns the intended blindspots
- Ensure that the intended blindspots are the only blindspots that the model learns

We clarify that the models that we trained using **synthetic data do not have "naturally occurring" blindspots** (i.e. blindspots beyond the known true blindspots that we induce).  For these models, we validated that the model has near-perfect (~99%+) accuracy on images that do not belong to the true blindspots, and near-zero accuracy on images belonging to the true blindspots.  This rules out the possibility of the existence of other coherent and underperforming subgroups.

**Point #3: Consistency between Synthetic vs. Real Data Experiments (Reviewer pnqW)**

Reviewers noticed two potential inconsistencies between our results on synthetic and real data.

*Observation 1*:  Spotlight has a higher average DR than Domino on synthetic data (Table 2), but Domino has a higher average DR than Spotlight on real data (Table 4).
- *Explanation*:  Domino uses CLIP’s representation which we hypothesize is better adapted to real data than to Synthetic data while Spotlight uses the model’s representation.  When we modified Domino to use the model’s embedding (leaving its core clustering methodology unchanged), Domino achieved performance competitive with Spotlight (average DRs 0.80 vs. 0.79) on the Synthetic data.

*Observation 2*:  On the real data, Domino has a higher average DR than PlaneSpot when there is a single less-specific blindspot (Figure 6 and 7), and PlaneSpot has a higher average DR than Domino overall (Table 4).  In contrast, PlaneSpot always has a higher average DR than Domino on the synthetic data.
- *Explanation*:  On the synthetic data, both PlaneSpot and Domino find 100% of the blindspots when there is a single less-specific blindspot in the model.  So the synthetic data does not provide a meaningful comparison between the methods for the real data result in question, and there is no inconsistency.
- *Aside*:  The fact that Domino is better than PlaneSpot in settings with a single blindspot and PlaneSpot is better in settings with 2+ blindspots is a good example of why identifying “experimental factors” that influence BDM performance is important, as for example [1] only studied settings with 1 blindspot.

Overall, we think that the results between these settings are remarkably consistent.

References:

[1] Sabri Eyuboglu, Maya Varma, Khaled Kamal Saab, Jean-Benoit Delbrouck, Christopher Lee-Messer, Jared Dunnmon, James Zou, and Christopher Re. Domino: Discovering systematic errors with cross-modal embeddings. In International Conference on Learning Representations, 2022. URLhttps://openreview.net/forum?id=FPCMqjI0jXN.

---

### Decision · Program_Chairs · 2023-01-20

**Decision:**

Reject

**Justification For Why Not Higher Score:**

The major concerns from reviewers are that experiments are synthetic. It will be more useful to demonstrate the proposed approach on real world experiments. It is not clear whether the trends on artificially induced blindspots are representative of the trends on naturally occurring blindspots.

**Justification For Why Not Lower Score:**

N/A

**Metareview: Summary, Strengths And Weaknesses:**

Summary; the authors studied the Blindspot Discovery Methods (BDMs) problem. The paper proposes SpotCheck framework for evaluating BDMs and a new BDM, PlaneSpot. The paper shows PlaneSpot outperforms existing BDMs.

Strength: the paper studies an important problem – providing a taxonomy and evaluation framework for the "blindspot detection" methods. The formalization is clear.

Weakness: the major concerns from reviewers are that experiments are synthetic. It will be more useful to demonstrate the proposed approach on real world experiments. It is not clear whether the trends on artificially induced blindspots are representative of the trends on naturally occurring blindspots.
Also note that authors should leverage the discussion period with reviewers rather than posting responses on the last day, which makes it hard for authors/reviewers/AC to discuss.